# Lysophosphatidic Acid Receptor 3 Activation Is Involved in the Regulation of Ferroptosis

**DOI:** 10.3390/ijms25042315

**Published:** 2024-02-15

**Authors:** Yi-Xun Huang, Kuan-Hung Lin, Jui-Chung Chiang, Wei-Min Chen, Hsinyu Lee

**Affiliations:** 1Department of Life Science, National Taiwan University, Taipei 10617, Taiwan; r10b21015@ntu.edu.tw; 2Institute of Plant and Microbial Biology, Academia Sinica, Taipei 115201, Taiwan; kuanhung@gate.sinica.edu.tw; 3Division of Molecular Radiation Biology, Department of Radiation Oncology, University of Texas Southwestern Medical Center at Dallas, Dallas, TX 75390, USA; jui-chung.chiang@utsouthwestern.edu

**Keywords:** lysophosphatidic acid, lysophosphatidic acid receptor 3, ferroptosis, erythropoiesis, lipid peroxidation, iron accumulation

## Abstract

Ferroptosis, a unique form of programmed cell death trigged by lipid peroxidation and iron accumulation, has been implicated in embryonic erythropoiesis and aging. Our previous research demonstrated that lysophosphatidic acid receptor 3 (LPA_3_) activation mitigated oxidative stress in progeria cells and accelerated the recovery of acute anemia in mice. Given that both processes involve iron metabolism, we hypothesized that LPA_3_ activation might mediate cellular ferroptosis. In this study, we used an LPA_3_ agonist, 1-Oleoyl-2-O-methyl-rac-glycerophosphothionate (OMPT), to activate LPA_3_ and examine its effects on the ferroptosis process. OMPT treatment elevated anti-ferroptosis gene protein expression, including solute carrier family 7 member 11 (SLC7A11), glutathione peroxidase 4 (GPX4), heme oxygenase-1 (HO-1), and ferritin heavy chain (FTH1), in erastin-induced cells. Furthermore, OMPT reduced lipid peroxidation and intracellular ferrous iron accumulation, as evidenced by C11 BODIPY™ 581/591 Lipid Peroxidation Sensor and FerroOrange staining. These observations were validated by applying *LPAR3* siRNA in the experiments mentioned above. In addition, the protein expression level of nuclear factor erythroid 2-related factor (NRF2), a key regulator of oxidative stress, was also enhanced in OMPT-treated cells. Lastly, we verified that LPA_3_ plays a critical role in erastin-induced ferroptotic human erythroleukemia K562 cells. OMPT rescued the erythropoiesis defect caused by erastin in K562 cells based on a *Gly A* promoter luciferase assay. Taken together, our findings suggest that LPA_3_ activation inhibits cell ferroptosis by suppressing lipid oxidation and iron accumulation, indicating that ferroptosis could potentially serve as a link among LPA_3_, erythropoiesis, and aging.

## 1. Introduction

Ferroptosis is a recently characterized programmed cell death process marked by the accumulation of iron and lipid peroxides, manifesting when the glutathione (GSH)-dependent lipid peroxide repair system is compromised [1,2]. The amino acid reverse transporter system Xc-, located in the plasma membrane, consists of two fundamental components: the light-chain subunit solute carrier family 7 member 11 (SLC7A11), and the heavy-chain subunit solute carrier family 3 member 2 (SLC3A2). This system regulates the balance of glutamate and cysteine by exchanging intracellular glutamate and extracellular cystine in a 1:1 ratio. Cystine is rapidly reduced to cysteine, a precursor essential for glutathione synthesis. On the other hand, GPX4 has been reported as a phospholipid hydroperoxidase [3], altering the state of phospholipid hydroperoxide from cytotoxic to non-toxic phospholipid alcohol. This prevents the formation and accumulation of lethal lipid reactive oxygen species (ROS) at the expense of GSH. Additionally, GPX4 exhibits antioxidant properties, shielding against cell membrane damage and lipid peroxidation during the ferroptosis process [4,5,6].

Iron is a crucial trace element in the human body, existing in two oxidation states: ferrous (Fe^2+^) and ferric (Fe^3+^). The iron redox cycle affects the sensitivity of cells to ferroptosis [7]. In its Fe^3+^ form, circulating iron is bound to transferrin and is imported into cells via the membrane protein transferrin receptor 1 (TFR1). Following absorption, STEAP3 metallo-reductase in the endosome reduces Fe^3+^ to Fe^2+^, which is then released into the cytosol through solute carrier family 11 member 2 (SLC11A2). Excess cellular iron ions are stored in ferritin, an intracellular iron storage protein complex comprising ferritin light chain (FTL) polymer and ferritin heavy chain 1 (FTH1). Additionally, ferroxidases can reoxidize Fe^2+^ to Fe^3+^, which is then exported into the extracellular space by the iron-efflux protein ferroportin (FPN) [8,9,10,11]. These events regulate cellular iron uptake and are pivotal for ferroptosis induction [12].

Lysophosphatidic acid (LPA) serves as a lipid mediator, modulating various physiological functions through the activation of six LPA G protein-coupled receptors, denoted as LPA_1_-LPA_6_ [13]. Our prior study revealed that lysophosphatidic acid receptor 3 (LPA_3_) mitigates oxidative stress and cellular senescence in Hutchinson-Gilford progeria syndrome (HGPS) [14]. The activation of LPA_3_ by the agonist OMPT (1-Oleoyl-2-O-methyl-rac-glycerophosphothionate) resulted in reduced reactive oxygen species levels in progerin HEK293 cells, achieved by upregulating the ROS scavenger genes NAD(P)H quinone dehydrogenase 1 (NQO1), NRF2, superoxide dismutase 2 (SOD2), and glutathione peroxidase 1 (GPX1). Moreover, OMPT restored phenylhydrazine (PHZ)-induced acute hemolytic anemia in mice [15]. These data strongly suggest that activating the LPA_3_ signaling pathway may play a role in aging-dependent anemia.

In progerin-expressing K562 cells, LPA_3_ protein expression levels were decreased [16,17], suggesting that LPA_3_ signaling is inhibited during aging. Conversely, the activation of LPA_3_ by OMPT resulted in the upregulation of erythrocyte markers, including glycophorin A (Gly A) and γ-globin. Interestingly, several studies indicated that red blood cell (RBC) numbers are decreased and that their function is impaired during the aging process due to a high level of oxidative stress [18,19,20]. These findings suggest that LPA_3_ may serves as an essential regulator both in preventing cellular aging by mitigating oxidative stress and by being a significant regulator during erythropoiesis.

Emerging research has demonstrated the relevance of ferroptosis in various physiological processes, including murine embryonic erythropoiesis and aging in different organs [21]. However, the relationship between LPA_3_ signaling and ferroptosis remains poorly understood. Therefore, the aim of this study was to investigate the impact of LPA_3_ on ferroptosis and to establish the potential link among LPA_3,_ erythropoiesis, and aging.

## 2. Results

### 2.1. An LPA_3_ Agonist Mitigates Erastin-Induced Ferroptotic Death

Erastin, as a ferroptosis inducer, acts by inhibiting system Xc- and reducing intracellular GPX4 levels [8]. To assess the toxicity of erastin in HT-1080 cells, varying erastin concentrations, ranging from 0 to 10 μM, were administered for 12 or 24 h. Cell viability was evaluated using the CCK-8 assay. The results demonstrated erastin-induced cytotoxicity in HT-1080 cells in a dose- and time-dependent manner (Figure 1A). The IC50 values were determined as 4.25 μM and 0.5549 μM at 12 and 24 h post-treatment, respectively.

We also assessed the levels of Fe^2+^, a trigger for Fenton’s reaction leading to lipid peroxidation. The fluorescence intensity from the FerroOrange Fluorescent probe was utilized to measure Fe^2+^ levels. HT-1080 cells were treated with erastin at various concentrations, ranging from 0 to 10 μM, for 12 or 24 h. The results revealed that erastin induced the dose- and time-dependent accumulation of Fe^2+^ in HT-1080 cells (Figure 1B). The iron accumulation exhibited a significant increase at 10 μM for 12 h and 5 μM for 24 h (*p*-value < 0.01). Therefore, we employed a concentration of 5 μM for a duration of 24 h in subsequent experiments.

To validate whether LPA_3_ activation affects cell viability in erastin-induced ferroptotic cell death, HT-1080 cells were treated with either vehicle or 5 μM erastin, in the absence or presence of 10 μM OMPT, for 24 h. Morphological observations revealed a distinctive “ballooning” phenotype indicative of ferroptosis in the erastin-treated cells. Notably, the addition of OMPT mitigated the erastin-induced ballooning phenotype (Figure 1C).

In our subsequent investigations, we sought to examine erastin-induced cell death in the absence of OMPT while concurrently treating with deferoxamine (DFO, an iron chelator) [22], N-acetylcysteine (NAC, antioxidant) [23], or ferrostatin-1 (Fer-1, ferroptosis inhibitor) [24]. The data showed that erastin treatment significantly reduced HT-1080 cell viability compared to that with vehicle treatment. However, co-treatment with OMPT, DFO, NAC, or Fer-1 effectively attenuated erastin-induced ferroptotic cell death. These findings suggest that the activation of LPA_3_ may play a crucial role in preventing erastin-induced ferroptosis (Figure 1D).

### 2.2. An LPA_3_ Agonist Mitigates Erastin-Induced Lipid Peroxidation in HT-1080 Cells

Excessive ROS can target polyunsaturated fatty acids in cellular or organ membranes, resulting in lipid peroxidation [25]. In our previous study, we proposed that lysophosphatidic acid (LPA) modulates reactive oxygen species (ROS) levels and cellular senescence via LPA_3_, attenuating cellular aging in Hutchinson-Gilford progeria syndrome (HGPS) cells [14]. Therefore, we aimed to validate the role of LPA_3_ in stabilizing lipid peroxidation in erastin-induced cells. To assess the lipid oxidation level, we applied C11 BODIPY 581/591, a widely used indicator of lipid oxidation. The results indicated that erastin treatment increased the level of lipid ROS (Figure 2A, Gray area). However, the activation of LPA_3_ mitigated erastin-induced lipid oxidation in HT-1080 cells (Figure 2A). Additionally, we investigated whether the activation of LPA_3_ regulates lipid ROS levels through intracellular antioxidants. We measured SLC7A11 and GPX4 protein expression levels, which are important free radical scavengers during ferroptosis. The results showed that erastin treatment reduces the protein expression level of SLC7A11 and GPX4. However, the activation of LPA_3_ rescued the expression level of SLC7A11 and GPX4 under erastin treatment (Figure 2B). To further address the role of LPA_3_ in ferroptosis, we utilized siRNA to knock down *LPAR3* in HT-1080 cells. The results showed that *LPAR3* knockdown decreased SLC7A11 and GPX4 protein and mRNA levels (Figure 2C,D). Together, these data suggest that the activation of LPA_3_ signaling protects cells from lipid peroxidation during erastin-induced ferroptosis.

### 2.3. An LPA_3_ Agonist Stabilizes Iron Homeostasis in Erastin-Induced HT-1080 Cells

Iron accumulation is also an important feature during ferroptosis. Next, we aimed to validate whether LPA_3_ is involved in iron accumulation by analyzing the Fe^2+^ level in erastin-induced cells. To determine this, we used the FerroOrange Fluorescent probe for detecting intracellular Fe^2+^ [26]. DFO, NAC, and Fer-1 served as inhibitors of ferroptosis and were used as positive controls. The results indicated that the Fe^2+^ level was significantly increased with erastin treatment. However, the co-treatment of erastin with OMPT resulted in a significant decrease of Fe^2+^ levels in erastin-induced cells (Figure 3A), suggesting that LPA_3_ activation reduces Fe^2+^ levels induced by erastin.

To further validate whether LPA_3_ activation regulates iron metabolism and storage function during ferroptosis, we measured the protein expression levels of HO-1 and FTH1, which have been reported to be involved in cellular iron homeostasis control [27]. Treatment with OMPT increased HO-1 and FTH1 protein expression levels in erastin-induced cells (Figure 3B), suggesting that LPA_3_ activation promotes iron metabolism. We also applied siRNA to knock down *LPAR3* in HT-1080 cells. The results showed that *LPAR3* knockdown led to the downregulation of HO-1 and FTH1 protein and mRNA levels (Figure 3C). These findings indicate that LPA_3_ has an important role in stabilizing Fe^2+^ homeostasis by regulating the expression level of iron metabolism/storage proteins and thereby affects ferroptosis in cells.

Our prior study revealed that LPA_3_ mitigates oxidative stress by upregulating NRF2 [14]. Furthermore, MEK-ERK activation has been reported to be one of the major pathways in the activation of NRF2 [28]. Thus, we proposed to explore whether LPA_3_ regulates ferroptosis via MEK/ERK/NRF2-dependent oxidative stress. Western blotting was applied to measure the phosphorylation level of p-MEK and p-ERK in the MAPK pathway, along with NRF2 protein levels, which has been reported as a regulator of ROS in HPGS during OMPT treatment [14]. The results showed that erastin-induced ferroptosis led to a reduction in NRF2 protein levels. However, treatment with OMPT significantly increased the protein levels of p-MEK, p-ERK, and NRF2 (Figure 3D). This indicates that erastin treatment effectively suppressed ROS production, but that LPA_3_ activation restored NRF2 levels. This recovery appears to be dependent on the MAPK pathway. Taken together, these results suggest that LPA_3_ might prevent cells from erastin-induced ferroptosis via the MEK/ERK/NRF2 signaling pathway.

### 2.4. LPA_3_ Promotes Erythropoiesis in Erastin-Induced K562 Cells

In our previous study, we demonstrated that LPA_3_ plays major roles in the regulation of myeloid differentiation [15]. Therefore, we aimed to validate whether LPA_3_-dependent ferroptosis is involved in the differentiation of erythroblasts. To achieve this, we initially established a human erythroleukemic cell line, K562, expressing *firefly* luciferase driven by the *GYPA* promoter. Cells were treated with OMPT in the absence or presence of erastin, while 200 nM of hemin was used as the positive control. Treatment with both hemin and OMPT, respectively, increased luciferase activity, indicating enhanced *GYPA* transcriptional activity. Conversely, treatment with erastin significantly decreased *GYPA* luciferase signals, suggesting that ferroptosis induction disrupts the differentiation of red blood cells (RBCs). However, when OMPT was applied along with erastin, it increased *Gly A* transcriptional activity under the erastin-treated conditions (Figure 4B), which was consistent with our previous observations [15].

To further investigate the roles of LPA_3_ in this signaling pathway, we used siRNA to knock down *LPAR3* in K562 cells. The results indicated that *LPAR3* knockdown decreased the transcription activity of *Gly A*. However, *Gly A* transcriptional activity did not respond to the treatment of OMPT (Figure 4C). Additionally, erastin treatment decreased *Gly A* gene expression both in control and LPA_3_-suppressed K562 cells (Figure 4D). Taken together, these data suggest that LPA_3_ may play an important role in regulating erythropoiesis in erastin-induced K562 cells.

## 3. Discussion

Ferroptosis, characterized by lipid peroxidation and iron accumulation, is a recently identified programmed cell death pathway. Our study reveals that activating LPA_3_ mitigates ferroptosis. Using the agonist OMPT, LPA_3_ activation elevated anti-ferroptosis gene expression, reducing lipid peroxidation and intracellular iron in erastin-induced cells. Additionally, LPA_3_ activation enhanced NRF2 expression, a key oxidative stress regulator. In human erythroleukemia K562 cells, LPA_3_ activation rescued erastin-induced ferroptosis and erythropoiesis defects. Overall, our findings suggest that LPA_3_ activation inhibits ferroptosis by suppressing lipid oxidation and iron accumulation, potentially linking LPA_3_ to erythropoiesis and aging.

Our results propose a potential mechanism by which LPA_3_ activation stabilizes intracellular Fe^2+^, a key factor contributing to ferroptosis [29]. While the reduction in intracellular Fe^2+^ levels upon OMPT treatment may not be as pronounced as with other known ferroptosis inhibitors, like DFO, NAC, and Fer-1, it still significantly attenuates Fe^2+^ accumulation compared to that in cells treated solely with erastin. This suggests that the LPA_3_ signaling pathway may be a damage response during ferroptosis.

The analyses of HO-1, pMEK, and pERK indicate that the combined treatment of OMPT and erastin produces a response similar to treatment with OMPT alone. This contrasts the responses observed for FTH1 and NRF2, where treatment with OMPT alone resulted in higher expression levels, as expected. This discrepancy suggests a likely indirect correlation among the variables under consideration. From our perspective, there is notable variability in outcomes with the co-treatment of OMPT and erastin across different groups, with distinct responses observed for HO-1, pMEK, and pERK compared to FTH1 and NRF2. However, when compared to exclusive treatment with erastin, each group subjected to co-treatment with OMPT and erastin demonstrates the restoration of approximately 20–30% of protein expression levels. Therefore, we propose that, in terms of efficacy, the outcomes show comparability across all groups. Nevertheless, these findings underscore the need for further investigation into the intricate molecular interactions involved in the LPA_3_-mediated inhibition of ferroptosis.

Previous studies have demonstrated that ferroptosis is involved in physiological processes, including embryonic erythropoiesis [30,31] and aging [32,33,34]. By using the HNEJ-1 mouse monoclonal antibody, which specifically recognizes HNE-derived Michael adducts expressed in histidine, lysine, and cysteine residues, ferroptosis events were monitored throughout the entire life span of rats [21]. A notable age-related increase in ferroptosis was observed in various organs, accompanied by iron accumulation. This phenomenon extended from embryonic stages to advanced age, implicating ferroptosis in both embryonic erythropoiesis and aging processes.

The prevalence of anemia In the elderly population is associated with declining hemoglobin levels [35]. The level of hemoglobin, which is composed of more than 60% of iron, significantly decreases with aging [36]. However, damaged erythrocytes under severe hemolytic anemia lead to more serious erythrophagocytosis by splenic red pulp macrophages (RPM) [30]. Similarly, patients with liver cirrhosis were shown to have lower serum and hepatic transferrin, resulting in iron overloading and ferroptosis in multiple organs [37,38,39]. Our recent findings on LPA_3_ activation restoring erythrocyte numbers under acute hemolytic anemia conditions suggest a potential therapeutic avenue for reducing systemic ferroptosis in aging-induced anemia [15]. To further investigate the connection between ferroptosis and erythropoiesis, we utilized a K562 cell model expressing firefly luciferase driven by the *Gly A* promoter, a marker of matured erythroid cells. Our results indicate that OMPT increases luciferase activity of the *Gly A* promoter during ferroptosis, supporting the link among LPA_3_ activation, ferroptosis, and erythropoiesis. Knockdown experiments further strengthen this connection, emphasizing the role of LPA_3_ in regulating these cellular processes.

In conclusion, our study substantiates that the activation of LPA_3_ signaling indeed regulates ferroptosis and erythropoiesis. This finding not only enhances our understanding of the intricate molecular mechanisms governing these cellular processes but also holds promise for potential therapeutic strategies targeting hematopoiesis disorders and age-related anemia.

## 4. Materials and Methods

### 4.1. Cell Culture and Pharmacological Reagents

HT-1080 human fibrosarcoma cells obtained from ATCC were cultured in Dulbecco’s modified Eagle’s medium (DMEM) supplemented with 10% fetal bovine serum (FBS) (Thermo Fisher Scientific Hyclone, Waltham, MA, USA), penicillin (100 U/mL), and streptomycin (100 U/mL) in a humid atmosphere containing 5% CO_2_ at 37 °C. K562 human erythroleukemia cells obtained from the ATCC were cultured in Roswell Park Memorial Institute (RPMI) 1640 medium supplemented with 10% fetal bovine serum (FBS) (Thermo Fisher Scientific Hyclone, Waltham, MA, USA), penicillin (100 U/mL), and streptomycin (100 U/mL) in a humid atmosphere containing 5% CO_2_ at 37 °C.

1-Oleoyl-2-O-methyl-rac-glycerophospothionate (OMPT, Cayman Chemicals, Ann Arbor, MI, USA), erastin (MilliporeSigma, Burlington, MA, USA), desferrioxamine mesylate (DFO, Abcam, Cambridge, UK), and ferrostatin-1 (Fer-1, Abcam) were dissolved in dimethylsulfoxide (DMSO, MilliporeSigma, Burlington, MA, USA). N-acetylcysteine (NAC, MilliporeSigma, Burlington, MA, USA) and hemin were dissolved in sterile water.

### 4.2. Cell Viability Assay

The cell viability assay was determined using the Cell Counting Kit-8 (CCK-8, Dojindo, Kumamoto, Japan). HT-1080 cells were seeded at 1000 cells per well in 96-well plates for 24 h. Cells were starved 4 h before reagent treatment. Next, the reagent was treated and incubated for 12 or 24 h. After reagent treatment, 10 μL of CCK-8 solution mixed with 90 μL of culture medium was added to each well, and the plates were incubated for an additional 1 h. Cell viability was evaluated by measuring the absorbance at a 450 nm wavelength using a SpectraMax i3X (Molecular Devices, San Jose, CA, USA).

### 4.3. Intracellular Ferrous Iron Content Detection

Intracellular ferrous iron content detection was determined using FerroOrange (Dojindo, Kumamoto, Japan). HT-1080 cells were seeded at a density of 1000 cells per well in 96-well plates for 24 h. Cells were starved 4 h before reagent treatment. Then, the reagent was treated and incubated for 12 or 24 h. Subsequently, reagent treatment was initiated, and cells were incubated for either 12 or 24 h. Following reagent treatment, the supernatant was discarded, and the cells were washed three times with DPBS. A 1 μM FerroOrange working solution with DPBS was then added to the cells, and they were incubated for 30 min. The intracellular ferrous iron content was quantified using a SpectraMax i3X (Molecular Devices), with excitation and emission bands set at 543 nm and 580 nm, respectively.

### 4.4. Measurement of Lipid ROS Level

The level of lipid reactive oxygen species (ROS) was quantified using the C11 BODIPY™ 581/591 Lipid Peroxidation Sensor (Thermo Fisher Scientific, Waltham, MA, USA). HT-1080 cells were seeded at a density of 50,000 cells per well in six-well plates, followed by incubation for 24 h. Cells were starved 4 h before reagent treatment. Subsequently, cells were treated with erastin (5 μM) in the presence or absence of OMPT (10 μM) for 24 h. Following reagent treatment, cells were washed with PBS. Cells were then stained with C11 BODIPY™ 581/591 for 20 min in a 37 °C incubator, followed by washing with PBS. Fluorescence signals were detected using the Guava^®^ Muse^®^ Cell Analyzer (Luminex Corp., Austin, TX, USA), with 10,000 cells measured and quantified for each experiment.

### 4.5. Western Blotting

Cells were washed with cold phosphate-buffered saline (PBS). Total protein from HT-1080 cells were extracted with RIPA buffer (150 mM NaCl, 1.0% NP-40, 0.5% sodium deoxycholate, 0.1% SDS, 50 mM Tris, pH 7.5) containing a 1% protease inhibitor cocktail (Merck Millipore, Billerica, MA, USA). Lysates were centrifuged at 4 °C at 14,000 rpm for 15 min. Next, supernatants were collected for cytometry proteins. Protein concentrations of cell lysates were quantified using the Pierce BCA protein assay kit (Thermo Scientific, Waltham, MA, USA). The samples were mixed with a 1/6 volume of 6x protein loading buffer with 2-ME and boiled at 100 °C for 10 min. The proteins were resolved via 10% SDS-polyacrylamide gel electrophoresis (90 V/30 min in 4% stacking gel, 130 V/1.5 h in a 10% running gel at 4 °C) and were transferred to polyvinylidene fluoride membranes (MilliporeSigma, Burlington, MA, USA, 100 V/90 min at 4 °C). The membranes were blocked with 5% bovine serum albumin (BSA, MilliporeSigma, Burlington, MA, USA) in TBST for one hour at room temperature. Subsequently, the membranes were exposed to primary antibodies, appropriately diluted in TBST supplemented with 1% BSA, at 4 °C overnight. Following this incubation, the membranes underwent three 10 min washes with TBST and were then incubated with a secondary antibody conjugated with horseradish peroxidase at room temperature for one hour. Then, membranes were mixed with ECL reagents (Advansta, Menlo Park, CA, USA) and imaged using a UVP ChemStudio Plus (Analytijena, Jena, Germany). The band intensities were then analyzed by VisionWorks. Antibodies used in this study were as follows: SLC7A11 (Abcam, ab37185), GPX4 (Abcam, ab125066), HO-1 (Abcam, ab13248), FTH1 (Abcam, ab65080), NRF2 (GeneTex, Irvine, CA, USA, GTX635826), MEK (Cell Signaling, Danvers, MA, USA, #9122), p-MEK (Cell Signaling, #9121), ERK (Cell Signaling, #4695), p-ERK (Cell Signaling, #9101).

### 4.6. RNA Interference and Transfection

*LPAR3* depletion was performed using the ON-TARGETplus siRNA SMARTpool (Dharmacon, L-004895-00-0005). The siRNAs were transfected using Lipofectamine^®^ 3000 (Invitrogen, Carlsbad, CA, USA) following the manufacture’s instruction. The siRNA sequences were as follows: non-target: 5′-UGGUUUACAUGUCGACUAA-3′, 5′-UGGUUUACAUGUUGUGUGA-3′, 5′-UGGUUUACAUGUUUUCUGA-3′, 5′-UGGUUUACAUGUUUUCCUA-3′; *LPAR3*: 5′-GGACACCCAUGAAGCUAAU-3′, 5′-UCUACUACCUGUUGGCUAA-3′, 5′-CAACACUGAUACUGUCGAU-3′, 5′-UCAUCAUGGUUGUGGUGUA-3′

### 4.7. Reverse Transcription (RT) and Real-Time Quantitative Polymerase Chain Reaction (RT-qPCR)

Total RNA was extracted from cells using TRI reagent (MilliporeSigma, Burlington, MA, USA). Complementary DNA (cDNA) was synthesized with 1 μg of total RNA using the Toyobo RT-PCR kit (Toyobo, Osaka, Japan). The iQ™ SYBR^®^ Green Supermix (Bio-Rad, Hercules, CA, USA) was utilized for quantitative real-time PCR assay in the Mini-Opticon real-time PCR system (Bio-Rad). Cycling conditions comprised an initial denaturation step at 95 °C for 5 min, followed by 40 cycles of denaturation at 95 °C for 30 s, annealing at 60 °C for 30 s, and extension at 72 °C for 10 min. To quantify target gene expression, each gene was normalized using *GAPDH* as an internal control. The specific primer sequences were as follows: *LPAR3*: forward (5′->3′): GAAGCTAATGAAGAC GGTGATGA, reverse (3′->5′): AGCAGGAACCACCTTTTCAC; *GAPDH*: forward (5′->3′): GGTGGTCTCCTCTGACTTCAAC, reverse (3′->5′): TCTCTCTTCCTCTT GTGCTCTTG

### 4.8. Luciferase Assay for GYPA Promoter Assay

The pGL4.32 [*luc2P/NF-κB-RE/Hygro*] plasmid (Promega, WI, USA) was utilized for luciferase assays. The luciferase plasmids were co-transfected with the *GYPA* promoter and *CMV-RFP*, serving as an internal control. The relative activities, calculated as the ratio of *Firefly* luciferase activity to *RFP* intensity, were quantified using the ONE-Glo™ Luciferase Assay system (Promega, WI, USA).

### 4.9. Statistical Analysis

The data are presented as the mean ± SD of a minimum of three independent experiments. Statistical analyses for multiple group comparisons of the mean ± SD were conducted using one-way analysis of variance (ANOVA). Significance levels were denoted as follows: * *p* < 0.05, ** *p* < 0.01, *** *p* < 0.001. Results with a *p*-value less than 0.05 were considered statistically significant.

## 5. Conclusions

In conclusion, our study establishes a crucial link between LPA_3_ activation and the regulation of ferroptosis, a distinctive programmed cell death process associated with lipid peroxidation and iron accumulation. Utilizing the LPA_3_ agonist OMPT, we demonstrated its potent impact on elevating anti-ferroptotic gene expression, reducing lipid peroxidation, and inhibiting intracellular ferrous iron accumulation in erastin-induced cells. Additionally, NRF2, a key oxidative stress regulator, was upregulated in OMPT-treated cells. Importantly, our findings highlight the critical role of LPA_3_ in mitigating ferroptosis in human erythroleukemia K562 cells, rescuing erastin-induced erythropoiesis defects. This study provides valuable insights into the potential therapeutic implications of LPA_3_ activation in modulating ferroptosis, offering a promising avenue for addressing aging-related and hematological disorders.

## Figures and Tables

**Figure 1 ijms-25-02315-f001:**
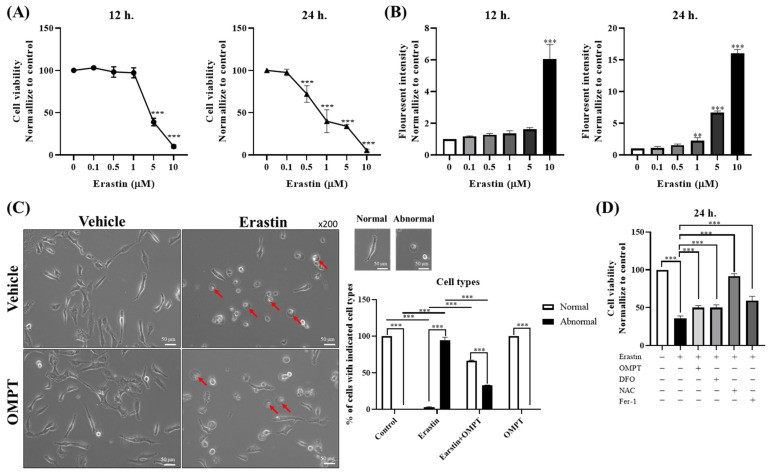
An LPA_3_ agonist mitigates erastin-induced ferroptotic death. (**A**) Cell viability of HT-1080 cells was measured via a CCK-8 assay after treatment with erastin at various concentrations (0 to 10 μM) for 12 or 24 h (*N* = 3). (**B**) Intracellular Fe^2+^ was measured using the FerroOrange Fluorescent probe after treatment with erastin at various concentrations (0 to 10 μM) for 12 or 24 h (*N* = 3). (**C**) HT-1080 cells were starved 4 h before reagent treatment and pre-treated with vehicle or erastin (5 μM) for 2 h, followed by treatment with or without OMPT (10 μM) for an additional 22 h (*N* = 3). Red arrows indicate cells undergo ferroptosis. (**D**) Cell viability was assessed by performing a CCK-8 assay. HT-1080 cells were starved 4 h before reagent treatment and pre-treated with vehicle or erastin (5 μM) for 2 h, followed by treatment with or without OMPT, DFO (200 μM), NAC (1 M), or Fer-1 (2 μM) for an additional 22 h (*N* = 3). Statistical analysis was performed using Student’s *t*-test and one-way ANOVA; ** *p* < 0.01, *** *p* < 0.001.

**Figure 2 ijms-25-02315-f002:**
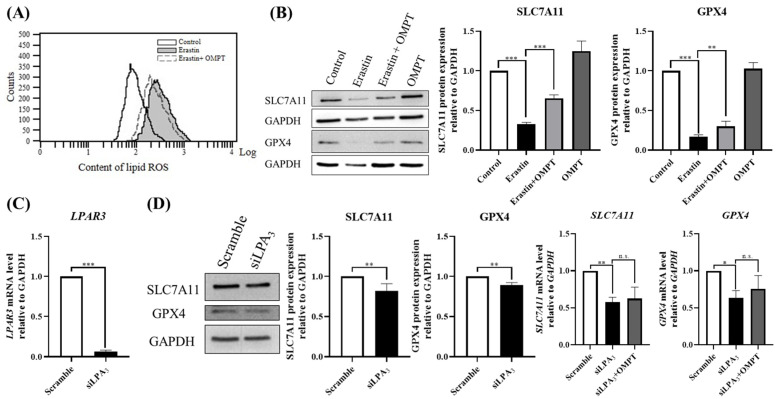
An LPA_3_ agonist mitigates erastin-induced lipid peroxidation in HT-1080 cells. (**A**) HT-1080 cells were starved 4 h before reagent treatment and pre-treated with erastin for 2 h, followed by treatment with or without OMPT for an additional 22 h. The fluorescence intensity of C11 BODIPY was measured via FACS, and the results showed that ROS levels decreased after OMPT treatment in erastin-induced cells. (**B**) HT-1080 cells were starved 4 h before reagent treatment and pre-treated with erastin for 2 h, followed by incubation with or without OMPT for 22 h. Western blots revealed that OMPT treatment increased the protein level of SCL7A11 (*N* = 3) and GPX4 (*N* = 3) in erastin-induced cells. (**C**) Real-time PCR results showed the knockdown of *LPAR3* mediated by siRNA (siLPA_3_) for 48 h (*N* = 3). (**D**) The knockdown of *LPAR3* mediated by siRNA (siLPA_3_) for 48 h decreased the protein level of SLC7A11 (*N* = 4) and GPX4 (*N* = 3) and mRNA (*N* = 3) levels. Statistical analysis was performed using one-way ANOVA and Student’s *t*-test; * *p* < 0.05, ** *p* < 0.01, *** *p* < 0.001, n.s. not significant.

**Figure 3 ijms-25-02315-f003:**
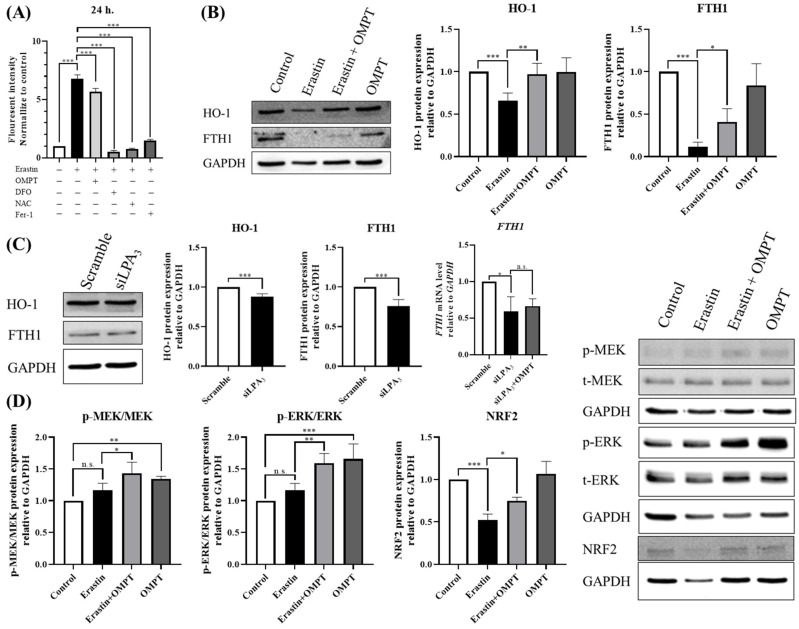
An LPA_3_ agonist stabilizes iron homeostasis in erastin-induced HT-1080 cells. (**A**) Intracellular Fe^2+^ was measured using the FerroOrange Fluorescent probe. HT-1080 cells were starved 4 h before reagent treatment and pre-treated with vehicle or erastin (5 μM) for 2 h, followed by treatment with or without OMPT, DFO (200 μM), NAC (1 M), or Fer-1 (2 μM) for 22 h (*N* = 3). (**B**) HT-1080 cells were starved 4 h before reagent treatment and pre-treated with erastin for 2 h, followed by treatment with or without OMPT for 22 h. Western blot analysis revealed that OMPT treatment increased the protein levels of HO-1 (*N* = 3) and FTH1 (*N* = 3) in erastin-induced cells. (**C**) The knockdown of *LPAR3* mediated by siRNA (siLPA_3_) for 48 h decreased HO-1 (*N* = 5) and FTH1 (*N* = 5) protein levels and the FTH1 (*N* = 3) mRNA level. (**D**) HT-1080 cells were starved 4 h before reagent treatment and pre-treated with erastin for 24 h, followed by treatment with or without OMPT for 5 min. Western blot analysis revealed that OMPT treatment increased p-MEK (*N* = 3) and p-ERK (*N* = 4) protein levels. HT-1080 cells were pre-treated with erastin for 2 h, followed by treatment with or without OMPT for an additional 22 h. Western blot analysis revealed that OMPT treatment increased the protein level of NRF2 (*N* = 4) in erastin-induced cells. Statistical analysis was performed using one-way ANOVA and Student’s *t*-test; * *p* < 0.05, ** *p* < 0.01, *** *p* < 0.001, n.s. not significant.

**Figure 4 ijms-25-02315-f004:**
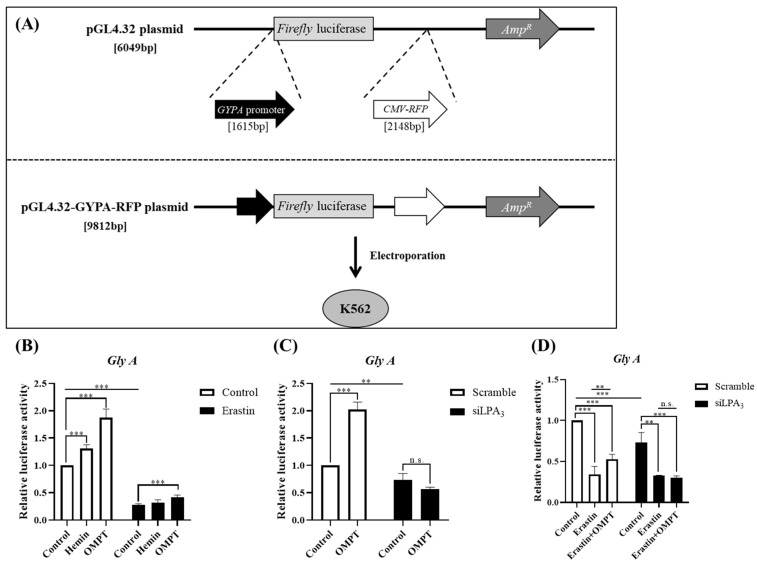
LPA_3_ promotes erythropoiesis in erastin-induced K562 cells. (**A**) Construction a luciferase-based screening system. (**B**) K562 cells were starved 4 h before reagent treatment and pre-treated with vehicle or erastin (5 μM) for 2 h, followed by treatment with or without hemin (200 nM) or OMPT (10 μM) for 22 h. The transcriptional activity of *Gly A* was decreased under erastin treatment, but was increased after OMPT treatment (*N* = 3). (**C**) Knockdown of *LPAR3* mediated by siRNA for 48 h resulted in a decreased expression level of *Gly A* (*N* = 4). (**D**) *LPAR3* knockdown mediated by siRNA lasted for 48 h, followed by starvation for 4 h before pre-treatment with erastin for 2 h, and then treatment with or without OMPT for 22 h in K562 cells. *Gly A* expression level did not respond to OMPT under *LPAR3*-knockdown conditions in K562 cells (*N* = 4). Statistical analysis was performed using one-way ANOVA; ** *p* < 0.01, *** *p* < 0.001, n.s. not significant.

## Data Availability

All relevant data are included in the manuscript and Appendix A.

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
