# Peer review of "Lysophosphatidic Acid Receptor 3 Activation Is Involved in the Regulation of Ferroptosis"

_ijms, 2024, doi:10.3390/ijms25042315_

Round 1

Reviewer 1 Report

Comments and Suggestions for Authors

This manuscript describes the role of LPA3 signaling in cell ferroptosis. The author found that activation of LPA3 inhibits ferroptosis by suppressing lipid oxidation and iron accumulation. While some novel findings were obtained in this study, there remain many concerns of its molecular mechanism and the research strategies which need to be addressed.

Major points

Although OMPT can potently activate the LPA3 receptor, it should be noted that OMPT is a pan-agonist that has agonist activity at other LPA receptors equal to or greater than that of LPA. (PMID: 23395664). In this study, the authors used OMPT at 10 uM, which is sufficient to activate LPA receptors other than LPA3. In addition, several LPA receptors are expressed in HT-1080 cells (PMID: 20484039, 30093116). Therefore, to confirm the dependence of LPA3 signaling in Figures 1-3, the authors need to show that the effect of OMPT is abolished in HT-1080 treated with LPAR3 siRNA, as in the K562 cells in Figure 4.

There appears to be little detection of GPX4 and FTH1 proteins by Western blotting in Erastin- and OMPT-treated cells. Therefore, another approach, such as improving this detection system or quantifying mRNA levels, would be needed to discuss the effects of OMPT treatment.

The authors argue that activation of LPA3 by OMPT ameliorates erastin-induced ferroptotic death, lipid peroxidation, and iron homeostasis, but the effect appears to be weak and partial. The authors use a medium with 10% serum, which is rich in ATX-dependently produced LPA and may be sufficient to induce LPA3 signaling. Thus, the role of LPA3 signaling may become more apparent when OMPT is stimulated under conditions where endogenous LPA3 signaling is depleted by the addition of ATX inhibitors.

Minor point

In Fig. 1C, please check if the % notation on the vertical axis is correct; is 1 a mistake for 100? Also, there is a misspelling (Nromal -> Normal)

Reviewer 2 Report

Comments and Suggestions for Authors

The article here presented by Huang et al. aims to dissect the role of Lysophosphatidic Acid Receptor 3 Activation in erythroid differentiation regulation through ferroptosis. 

The research design must be extensively revised. Firstly, the majority of the paper focuses on experiments with epithelial cells, while only the last paragraph addresses erythropoiesis. More importantly, the results on erythropoiesis appear to be preliminary and unclear. It would be better to completely change the focus of the article, taking erythropoiesis out of the paper unless performing further experiments. 

Major revisions. 

In figure 2D, the authors show western blot analyses of SLC7A11 and GPX4 , Images and histograms do not show any reduction compared to scrumble and I have doubts about the statistical significance. As in Figure 2D changes in protein expression are clear, I would suggest repeating this analysis at shorter time points. Moreover, I would also suggest testing gene expression levels to test whether LPA3-dependent regulation affects protein stability or gene expression. 

In figure 3A, histograms show Fe2+ concentration after erastin treatment plus OMPT, DFO NAC, and Fer-1 but no comments can be found in the corresponding paragraph of the results. Indeed, in this paragraph, the authors explained that the combined treatment of Erastin and OMPT resulted in a significant decrease of Fe2+ but this is more significant when cells are treated with Erastin and the other compounds (without OMPT). How do the authors explain this? 

In Figure 3C, I have the same concerns about the p-value as in Figure 2D.

In Figure 3D, the western blot of NRF2 cannot be considered informative as the quality is really poor. I am doubtful about the method the authors used for densitometric analysis, considering that, besides the absence of signal in samples 2 and 3, the GAPDH of the samples is also reduced. 

Finally, the analyses of HO-1, as well as pMEK and pERK, show that the treatment with OMPT and Erastin has a similar effect to OMPT alone (differently from FTH1 and NRF2 where the OMPT treatment alone induced higher expression levels, as expected). Probably, this suggests an indirect correlation of the considered variables, which is not a problem itself but should be discussed.  

As mentioned before, I would eliminate the last paragraph for several reasons. First of all, the experimental design is very weak, starting from the chosen cellular model and considering the strategy used for monitoring the erythroid differentiation. The erythroid focus is my major concern in this article. Once again, I suggest focusing on the characterization of the general role of lysophosphatidic Acid Receptor 3 activation. 

In conclusion, the Discussion section should be completely rewritten, as it currently reads more like an introduction than a concluding paragraph.

Minor revision

No error bars are shown for control and/or scramble in each histogram. 

The number of replicates must be specified for each experiment

Comments on the Quality of English Language

The overall quality of the English Language is acceptable. 

Round 2

Reviewer 1 Report

Comments and Suggestions for Authors

The results of the additional experiments (Supplementary Figure) are important and I would suggest that they be included in the Results section.

Author Response

Thank you for your suggestion. We have incorporated the results into Figure 2D and 3C, and provided corresponding explanations in the figure legends.

Reviewer 2 Report

Comments and Suggestions for Authors

The authors properly revised the manuscript and answered most of the comments. 

Author Response

Thank you very much for taking the time to review this manuscript.